

# Perception of the risk of adverse reactions to analgesics: differences between medical students and residents

Sandra Castillo-Guzman[1], Omar González-Santiago[2], Ismael A. Delgado-Leal[1], Gerardo E. Lozano-Luévano[1], Misael J. Reyes-Rodríguez[1], César V. Elizondo-Solis[1], Teresa A. Nava-Obregón[1] and Dionicio Palacios-Ríos[1]

[1] Pain and Palliative Care Clinic, Anesthesiology Service, University Hospital Dr Jose E Gonzalez, Universidad Autonoma de Nuevo Leon, Monterrey, Nuevo León, Mexico
[2] Posgraduate Division of the Faculty of Chemical Science, Universidad Autonoma de Nuevo Leon, Monterrey, Nuevo León, Mexico

## ABSTRACT

**Background.** Medications are not exempt from adverse drug reactions (ADR) and how the physician perceives the risk of prescription drugs could influence their availability to report ADR and their prescription behavior.

**Methods.** We assess the perception of risk and the perception of ADR associated with COX2-Inbitors, paracetamol, NSAIDs, and morphine in medical students and residents of northeast of Mexico.

**Results.** The analgesic with the highest risk perception in both group of students was morphine, while the drug with the least risk perceived was paracetamol. Addiction and gastrointestinal bleeding were the ADR with the highest score for morphine and NSAIDs respectively.

**Discussion.** Our findings show that medical students give higher risk scores than residents toward risk due to analgesics. Continuing training and informing physicians about ADRs is necessary since the lack of training is known to induce inadequate use of drugs.

Corresponding author
Sandra Castillo-Guzman, castilloguzsan@yahoo.com.mx

## INTRODUCTION

Analgesics are the cornerstone of pain management and their availability is critical for to alleviate unnecessary chronic and acute pain, especially in developing countries (*Lohman, Schleifer & Amon, 2010*). However, these medications are not exempt from adverse reactions (ADR). The use of opioids is associated with various ADRs ranging from nausea and vomiting to urinary retention and respiratory depression. Paracetamol is relatively safe when taken in a therapeutic dose ($\leq 4$ g/day for adults). However, overdosage leads to hepatotoxicity and nephrototoxicity (*Chun et al., 2009*; *Waring, Jamie & Leggett, 2010*; *Hodgman & Garrard, 2012*). Non-Steroidal Anti-inflammatory-drugs (NSAIDs) can result in gastrointestinal (GI) complications, ranging from dyspepsia to peptic ulcer and GI bleeding (*Castellsague et al., 2012*). On the other hand, COX2 inhibitors could create an

ulcerogenic dual-COX inhibitor when administered with low-dose aspirin. Moreover, by inhibiting COX2, they could delay ulcer healing. Similar to traditional NSAIDs, COX2 inhibitors compromise the glomerular filtration rate in patients at increased risk, and may cause peripheral edema and hypertension. In combination with an oral anticoagulant they increase the international normalized ratio (*Mattia & Coluzzi, 2005*).

On the other hand, how the physician perceives the risk of prescription drugs could influence their availability to report ADR and their prescription behavior. With opioids, an apprehensive attitude when using morphine as an analgesic could lead to resistance to administer morphine to patients suffering from severe pain. Such reluctance can have a negative impact on pain management as well as quality of life (*Joranson et al., 2000*; *Bandieri et al., 2009*).

With this in mind, the aims of this study was (1) to assess the risk perception of medical students (MS) and residents (Rs) towards the normal use of opioid and non-opioid analgesics, and (2) to assess the perception of common ADR caused by morphine and NSAIDs.

## METHODS

This study was conducted in the Faculty of Medicine of the Autonomous University of Nuevo León (UANL) and the Dr José E. Gonzalez University Hospital, both located in the Metropolitan area of Monterrey, Mexico. The sample of MS was conformed by those who had already taken a pharmacology course and were surveyed in the faculty of medicine (halls, study areas, library). The sample of Rs include those of any specialty and year of residence and were surveyed in the hospital.

After informing the aim of the study and obtaining verbal consent of the participants, the survey was applied. Participants were informed that the first section was optional. The questionnaire was self-administered with supervision.

### Instrument

The instrument was composed of three sections: the first section is general questions about gender, age and year of study or year of residence, as appropriate. This section was optional; the second section evaluated risk perception to analgesics when normal dosing was used; it included COX2-inhibitors, paracetamol, morphine and NSAIDs. This was performed as previously reported by *Durrieu et al.* (*2007*) and *Durrieu et al.* (*2010*). A visual analogue scale of 10 cm ranging from 0 (drug without risk) to 10 (highly risky drug) was used to assess perceived risk to each one of the analgesics previously mentioned. The value was obtained by measuring the distance between the left side of the scale (equal to zero) and the mark made by the participant. Since each scale measured 10 cm, risk perception could be considered a quantitative score ranging from 0 to 10. The third section evaluated risk perception to specific ADR when normal use of morphine and NSAIDs were used. The ADR evaluated were gastrointestinal (GI) bleeding, kidney damage, liver damage, sedation, bronchospasm, and addiction. Each adverse effect was assessed as a risk perception; that is, a visual analogue scale of 10 cm ranging from 0 (absent effect) to 10 (effect very frequent). Morphine was chosen due to the fact that it is a better-known opioid; therefore, it is a good

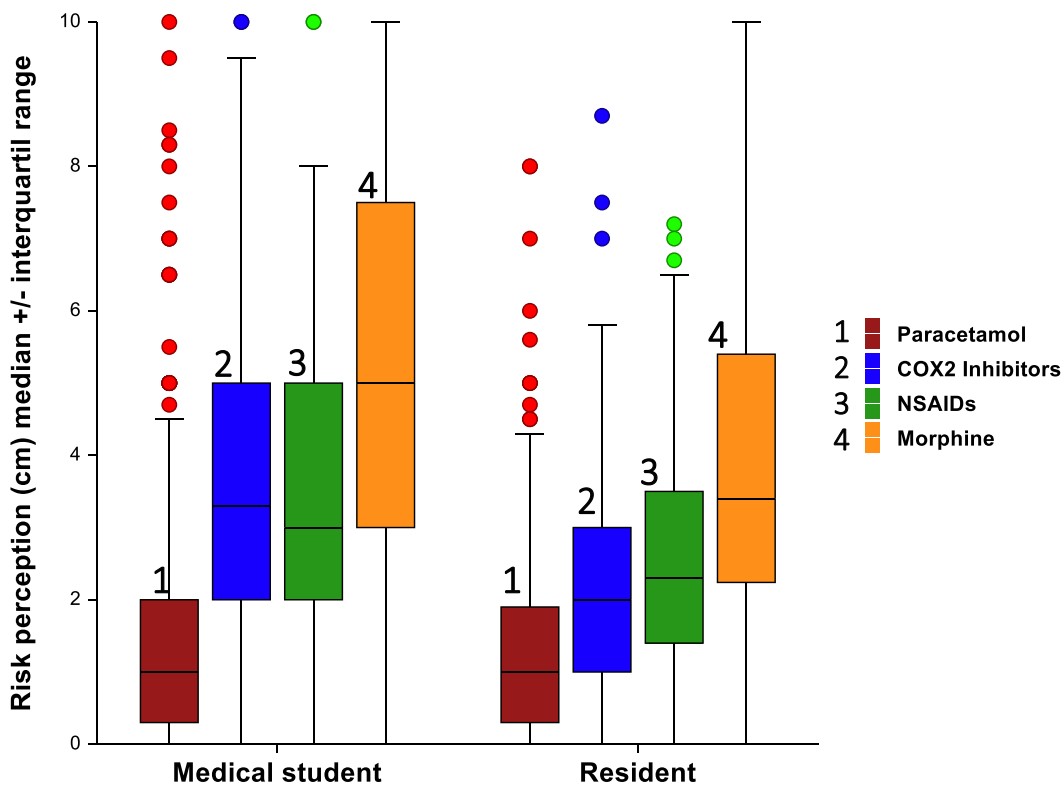

**Figure 1  Risk perception toward different analgesic between medical students and residents.**

representative of opioid analgesics. NSAIDs were chosen due to the fact that they are very familiar drugs to all students and they are a good representation of non-opioid analgesics.

### Statistical analysis

Normality of data was tested with the Kolmogorov–Smirnov test. Data are reported as median and 25th–75th percentiles. The Mann–Whitney $U$-test was used for comparison between two groups of students and between males and females. The statistical package SPSS V20 and NCSS-10 were used for all analyses.

### Ethical approval and consent

The Ethics committee of the Faculty of Medicine of the Autonomous University of Nuevo León approved this study and exempted from written informed consent. The reference number is AN15-011.

## RESULTS

Five hundred and five students were interviewed. Women and men represented 39.7% and 60.3%, respectively. MS on the other hand, represented 58.9% and Rs 41.1%.

### Risk perception to analgesics

Overall, the analgesic with the highest risk perception was morphine, while the drug with the least risk perceived was paracetamol (Fig. 1). This pattern was observed in MS and Rs of all
**Table 1  Risk perception to analgesics by year of study (median and interquartile range).**

| Degree | Year | Paracetamol | COX2 inhibitors | NSAIDs | Morphine |
|---|---|---|---|---|---|
| Medical students | 3 | 1 (0.3–2.5) | 4 (2.5–5) | 2.3 (1.5–4) | 6 (3.5–8.5) |
| | 4 | 1.3 (0.5–2) | 3 (2.5–5) | 3 (2.2–5) | 6 (4–7.5) |
| | 5 | 0.8 (0.4–2) | 3.7 (2.5–5.3) | 3.5 (2–5) | 5.5 (2.7–8) |
| | 6 | 1 (0–2) | 3 (1.5–5) | 3 (1.7–5) | 4 (2–6) |
| | *P* value | 0.25 | 0.09 | 0.03 | 0.00 |
| Residents | 1 | 1 (0.3–1.7) | 1.8 (0.9–3.1) | 2.3 (1.2–3.7) | 3.8 (2.5–5.3) |
| | 2 | 1 (0.3–1.8) | 2 (1–3) | 2.5 (1.3–3.8) | 4 (1.8–6.5) |
| | 3 | 1.5 (0.7–2) | 2 (1–3.4) | 2.5 (1.7–3.5) | 3 (2.2–5.4) |
| | 4 | 1.2 (0–1.8) | 2 (1.1–2.5) | 2 (1.4–3.5) | 3.3 (1.8–5.2) |
| | *P* value | 0.4 | 0.9 | 0.9 | 0.8 |

**Table 2  Risk perception to Analgesics between medical students and residents[a] according their gender (median and interquartile range).**

| | COX2 inhibitors | | | Paracetamol | | | Morphine | | | NSAIDs | | |
|---|---|---|---|---|---|---|---|---|---|---|---|---|
| | MS (n = 300) | R (n = 209) | P | MS (n = 300) | R (n = 209) | P | MS (n = 300) | R (n = 209) | P | MS (n = 300) | R (n = 209) | P |
| Total (N = 509) | 3.3 (3–3.7) | 2.0 (1.7–2) | 0.00 | 1.0 (1–1) | 1.0 (1–1.2) | 0.70 | 5.0 (5–6) | 3.4 (3.1–4) | 0.00 | 3 (2.7–3.3) | 2.3 (2–2.5) | 0.00 |
| Male (N = 307) | 3.5 (3–4) | 2.0 (1.7–2.5) | 0.00 | 1.0 (1–1.3) | 1 (0.8–1.2) | 0.29 | 5.26 (2.64) | 3.95 (2.31) | 0.00 | 3.0 (2.8–3.5) | 2.5 (2–2.7) | 0.00 |
| Female (N = 202) | 3 (2.7–3.8) | 1.8 (1.2–2) | 0.00 | 1.0 (0.5–1) | 1.0 (0.7–1.5) | 0.52 | 5.0 (4.5–6.2) | 3.4 (2.9–4) | 0.00 | 2.8 (2.3–3.3) | 2.2 (2–2.6) | 0.02 |
| *P* value | 0.10 | 0.12 | | 0.04 | 0.57 | | >0.05 | 0.91 | | 0.19 | 0.50 | |

**Notes.**
MS, Medical students; R, Residents; *P*, <0.05; S, Significant; NS, Nonsignificant.
[a]Comparison between MS and R was with Mann–Whitney test.

years of study (Table 1). In the case of MS, there was a significant difference among years of study and perception of risk to morphine and NSAIDs. For morphine, the higher risk score was observed in 3rd year medical students while the lowest score was in 6th year medical students. For NSAIDS, this was the opposite. With respect to Rs, there no was a significant difference between year of residence and risk perception score for each one analgesic.

MS had greater scores in risk perception than Rs (Table 2). This difference was significant for COX2 inhibitors, morphine and NSAIDs. These high scores were also observed in both genders of MS. Significant differences between females and males was observed only for paracetamol in MS ($P = 0.04$) (Table 2).

## Risk perception to ADR

For morphine, addiction and GI bleeding were the ADR with the highest and lowest scores, respectively in MS (8 and 5 respectively) and Rs (6.2 and 2.5 respectively). This pattern was observed in all years of study of both group of students (Table 3). There no were a significant difference among years of study of MS and risk perception of ADR except for bronchospasm ($P = 0.02$). In the case of Rs, there no were a significant difference among year of residence

Castillo-Guzman et al. (2016), *PeerJ*, DOI 10.7717/peerj.2255

**Table 3  Risk perception of adverse drug reactions due to morphine and NSAIDS by year of study.**

| Degree | Year | Morphine | | | | | | NSAIDS | | | | | |
|---|---|---|---|---|---|---|---|---|---|---|---|---|---|
| | | GI bleeding | Kidney damage | Liver injury | Sedation | Broncho espasma | Addiction | GI bleeding | Kidney damge | Liver injury | Sedation | Broncho espasma | Addiction |
| Medical students | 3 | 5 (3–7) | 5 (3–6.5) | 5 (2.8–7.5) | 8.5 (6.7–9.3) | 5.5 (3.5–7) | 8.7 (7–9.6) | 7.7 (6.5–9) | 6.5 (5–7.7) | 6.5 (5–8) | 3.7 (1.5–5.7) | 3.3 (1.5–5.7) | 2.7 (1.3–5) |
| | 4 | 5 (2.7–5.3) | 5 (3.5–6.5) | 5.7 (4.3–6.7) | 7.3 (6–8.7) | 6 (5–7.5) | 8 (6–9) | 7.5 (6–8.6) | 6 (5–7.6) | 6 (4–7.7) | 3.3 (2–5) | 3 (2–5) | 2.5 (1–6) |
| | 5 | 4.5 (2.5–6) | 5 (3.5–6.7) | 5 (3.5–6.7) | 7.5 (5.5–9) | 6 (3.7–7.3) | 8 (6.5–9.5) | 8 (5.7–9) | 7 (5–8.5) | 7 (4.3–8.1) | 4.5 (2–6.5) | 3.2 (1.7–5) | 2.5 (1–5) |
| | 6 | 4.2 (2–6) | 4.1 (2–6.4) | 5 (2.7–6.3) | 8 (7–9) | 5 (3–7) | 8.5 (7–9.7) | 8 (6.5–9) | 7 (5–8.5) | 6.5 (4.7–9) | 3 (1.5–5) | 3 (1.5–5) | 2 (1–4) |
| | P value | 0.4 | 0.21 | 0.11 | 0.08 | 0.02 | 0.18 | 0.43 | 0.03 | 0.61 | 0.22 | 0.76 | 0.57 |
| Residents | 1 | 2.2 (1.2–3.8) | 2.9 (1.9–4.8) | 4.2 (2.2–5.5) | 6.3 (4–8.1) | 4.5 (2.4–5.4) | 6.6 (5.4–8.1) | 7.1 (5.7–8.2) | 6 (4.5–7.5) | 4.7 (2.2–6.1) | 2 (1.2–3.2) | 2 (1.4–2.9) | 2.2 (1.2–3) |
| | 2 | 2.4 (1–4) | 3 (1.4–4.8) | 3 (1.9–5) | 6.3 (3.6–8) | 3.2 (1.7–5.5) | 5.8 (3.1–7.4) | 6.5 (4.3–8.3) | 6.1 (2.7–7) | 3.6 (1.5–6) | 1.7 (0.3–2.8) | 2 (1–3) | 1.7 (0.7–2.8) |
| | 3 | 2.6 (1–3.3) | 3 (1.8–5.4) | 3.8 (2–5.8) | 6.2 (4–7.6) | 4.3 (2–6.6) | 6 (4.4–8.2) | 7.2 (5.7–8.7) | 6.8 (4.3–8) | 4.8 (1.5–6.5) | 1.5 (0.7–3.2) | 2.1 (1–3.5) | 2 (0.8–3.3) |
| | 4 | 2.6 (1.6–4) | 3.8 (2.4–5.6) | 4.1 (3–5.7) | 6.5 (5.7–7.6) | 3.5 (2–5.8) | 7 (5.4–7.4) | 7 (6.4–8) | 6.7 (5.1–7.8) | 5.3 (2.9–7) | 2.6 (1.7–3.3) | 2.5 (1.8–3.2) | 2.5 (1.5–3.2) |
| | P value | 0.6 | 0.3 | 0.6 | 0.8 | 0.7 | 0.3 | 0.5 | 0.3 | 0.1 | 0.08 | 0.4 | 0.2 |

and risk perception of ADR due to morphine. On the other hand, MS showed significant higher scores than residents in the risk perception of ADR due to morphine (Table 3); this was similar in both genders of this group of students (Table 4). Significant differences between genders were observed in GI bleeding and sedation in the group of MS and Kidney damage in the case of residents.

For NSAIDs, the ADR with the highest score in both MS and residents, was GI bleeding (7.7 and 7.1, respectively), while the lowest score was addiction (2.5 and 2 respectively). In the case of MS, there no was a significant difference between year of study and risk perception to ADR, except for kidney damage ($P = 0.03$). In residents, there was no significant difference between year of residence and risk perception to ADR. On the other hand, MS showed significant higher scores than residents in the ADR due to NSAIDS except in kidney damage ($P = 0.06$). There no were significant difference between MS and Rs according gender in Kidney damage. Also, in the case of GI bleeding, there no were a significant difference between both groups of students of the female gender ($P = 0.09$). Finally, there were a significant difference in GI bleeding and sedation between males and females of the group of Rs.

## DISCUSSION

Previous studies have showed risk perception of drugs in health professionals, students and patients (*Durrieu et al., 2007*; *Durrieu et al., 2010*; *Cullen, Kelly & Murray, 2006*; *Bongard et al., 2002*). However, differences between medical students and residents has been poorly studied. In addition, studies have focus in several drug class and rarely in a specific drug. In this study, we searched for differences in the risk perception to the analgesics paracetamol, COX2 inhibitors, morphine, and NSAIDs between medical students and residents. Our findings show that medical students give higher risk scores than residents to all analgesics studied except paracetamol. This was independent of gender. We speculate that this difference could be explained by the recent courses of pharmacology taken by medical students. As has been previously showed, the pharmacology course increases global perception of risk (*Durrieu et al., 2007*; *Durrieu et al., 2010*). Others factors, such as the persuasive methods of pharmaceutical representatives, could affect these perceptions especially in residents who are more in touch with them than medical students.

In both groups of students, the decreased order of risk perception was as follows: morphine, NSAIDS, COX2 inhibitors and finally, paracetamol. There no were a significant difference between medical students and residents in this latter analgesic. We think that the low risk perceived for paracetamol, when normal doses are used, could have serious implications. It is probable that participants could be underestimating its potential hepatic risk. Previous studies have shown that paracetamol even at normal doses can produce liver injury (*Sabate, Ibañez & Pérez, 2011*). On the other hand, the highest risk score assigned to morphine by MS and R could suggest morphinofobia. More studies in this respect are necessary in Mexican physicians since with our data it is not possible to conclude. The term morphibophobia can be defined as either a number of beliefs based on the side effects of morphine prescribed for pain management, or an inadequate management of chronic pain due to lack of knowledge on how to use morphine (*Ferreira et al., 2013*).

Castillo-Guzman et al. (2016), *PeerJ*, DOI 10.7717/peerj.2255

**Table 4  Risk perception of adverse drug reactions due to morphine and NSAIDS between medical students and residents.**

| Drugs | GI bleeding | | | Kidney damage | | | Liver damage | | | Sedation | | | Bronchospasm | | | Addiction | | |
|---|---|---|---|---|---|---|---|---|---|---|---|---|---|---|---|---|---|---|
| | MS (*n* = 300) | R (*n* = 209) | P | MS (*n* = 300) | R (*n* = 209) | P | MS (*n* = 300) | R (*n* = 209) | P | MS (*n* = 300) | R (*n* = 209) | P | MS (*n* = 300) | R (*n* = 209) | P | MS (*n* = 300) | R (*n* = 209) | P |
| **Morphine** | | | | | | | | | | | | | | | | | | |
| Total (N = 509) | 5.0 (4.3–5) | 2.5 (2–2.7) | 0.00 | 5.0 (5–5) | 3.1 (2.7–3.4) | 0.00 | 5.0 (5–5.4) | 3.6 (3–4) | 0.00 | 8.0 (7.5–8) | 6.3 (6–6.8) | 0.00 | 5.5 (5–6) | 4 (3.2–4.5) | 0.00 | 8.0 (8–8.5) | 6.2 (5.8–6.7) | 0.00 |
| Male (N = 307) | 4.5 (4–5) | 2.4 (2–2.8) | 0.00 | 5.0 (4.7–5) | 2.8 (2.5–3.2) | 0.00 | 5.0 (5–5) | 3.3 (2.9–4) | 0.00 | 7.5 (7–8) | 6.4 (6–7) | 0.00 | 5.5 (5–6) | 4.2 (3–4.7) | 0.00 | 8.0 (8–8.5) | 6.5 (6–6.8) | 0.00 |
| Female (N = 202) | 5.0 (4.5–5.3) | 2.5 (2–3) | 0.00 | 5.0 (5–5.5) | 3.5 (3.1–4.3) | 0.00 | 5.0 (5–6) | 3.8 (3–4.8) | 0.00 | 8.3 (8–8.5) | 6.3 (5.7–7) | 0.00 | 5.5 (5–6) | 4.0 (2.7–4.7) | 0.00 | 8.5 (8–9) | 6.0 (5–7) | 0.00 |
| P value | 0.02 | 0.88 | | 0.14 | 0.04 | | 0.14 | 0.37 | | 0.02 | 0.43 | | 0.84 | 0.76 | | 0.10 | 0.52 | |
| **NSAIDs** | | | | | | | | | | | | | | | | | | |
| Total (N = 509) | 7.7 (7.5–8) | 7.1 (6.7–7.3) | 0.00 | 6.5 (6–7) | 6.3 (5.9–6.8) | 0.06 | 6.5 (6–7) | 4.5 (3.4–4.8) | 0.00 | 3.5 (3–4) | 2.0 (1.5–2.2) | 0.00 | 3.1 (3–3.7) | 2.0 (2–2.3) | 0.00 | 2.5 (2–3) | 2.0 (1.5–2.2) | 0.00 |
| Male (N = 307) | 7.5 (7–8) | 7.0 (6.4–7.2) | 0.00 | 6.5 (6–7) | 6.2 (5.7–6.8) | 0.26 | 6.5 (6–7) | 3.7 (3.1–4.7) | 0.00 | 3.5 (3–4.2) | 1.65 (1.3–2) | 0.00 | 3.3 (3–4) | 2.0 (1.5–2.4) | 0.00 | 2.5 (2–3) | 1.7 (1.5–2.2) | 0.00 |
| Female (N = 202) | 8.0 (7.3–8.7) | 7.3 (6.9–7.8) | 0.09 | 7.0 (6.5–7.5) | 6.5 (6–7) | 0.10 | 6.3 (5.5–7) | 5.0 (3.7–5.8) | 0.00 | 3.3 (3–4) | 2.3 (1.8–3) | 0.00 | 3.0 (2.7–4) | 2.3 (2–2.5) | 0.00 | 2.5 (2–3) | 2.2 (1.6–2.5) | 0.04 |
| P value | 0.18 | 0.04 | | 0.12 | 0.63 | | 0.94 | 0.10 | | 0.91 | 0.00 | | 0.50 | 0.14 | | 0.44 | 0.52 | |

Until now, there are no studies that report risk perception to paracetamol using the instrument of this study; therefore, it is not possible establish a comparison. The same applies to COX2 inhibitors and morphine. However, regarding NSAIDS, others studies have shown results different from this work in medical students who have taken a pharmacology course (median ≅ 6.3) (*Durrieu et al., 2007*) and medical staff (median 6.2) (*Bongard et al., 2002*).

With regard to perception of ADRs studied, both groups of students scored some level of risk perception of GI bleeding, kidney damage, liver injury, sedation, bronchospasm and addiction due to morphine and NSAIDs. GI bleeding had the highest score for NSAIDs, while addiction had the highest score for morphine. As with paracetamol, COX2 inhibitors and morphine, there are no studies similar to this study that allow us to compare the score of GI bleeding and addiction by NSAIDs and morphine, respectively, in health professionals. However in patients with regular use (*Cullen, Kelly & Murray, 2006*), the score reported (median = 3.8) is higher than our population of medical students and residents. Interestingly, women had higher risk scores than men in some ADR, and the causes of this are not clear. There are no similar studies that permit a comparison with other populations; however, it has been reported that women have major risk perception in other areas such as financial, health/safety, recreational, ethical, and social decisions (*Harries & Jenkins, 2006*). Finally, both groups of students score ADRs that are infrequent in NSAIDs and morphine, such as GI bleeding and addiction, respectively. The reasons or implications of this observation are not clear, but merit more research.

Since risk, perception to prescription drugs could influence the availability to report ADR and their prescription behavior, continuing training and informing physicians about ADRs is an important issue. In addition, the lack of training is known to induce inadequate use of drugs (*McDowell, Ferner & Ferner, 2009*). Poor training also could complicate the transmission of information to patients regarding ADRs, studies suggest the increase of information to patients results in a reduction of ADR, hospital admissions, morbidity and cost.

Finally, limitations of this study should be considered, these include the lack of randomization in the order of items in both "risk perception to analgesics" and "risk perception of ADR." Moreover, the risk perception to morphine may not be the same for all opioids. In addition, our results may not apply to other populations because other studies are probably different from our sample.

## CONCLUSIONS

There is a difference in the risk perception to analgesics between medical students and residents. The former have a higher risk score toward analgesics than the latter, except for paracetamol. In both groups of students, the decreasing risk was as follows: morphine, NSAIDs, COX2 inhibitor and paracetamol. GI bleeding and addiction were the ADRs with the highest score for NSAIDs and morphine, respectively in both groups.

## ACKNOWLEDGEMENTS

We thank Sergio Lozano-Rodriguez, M.D. for his help in editing the manuscript.

### Funding

The authors received no funding for this work.

### Competing Interests

The authors declare there are no competing interests.

### Author Contributions

- Sandra Castillo-Guzman conceived and designed the experiments, analyzed the data, contributed reagents/materials/analysis tools, wrote the paper, reviewed drafts of the paper.
- Omar González-Santiago conceived and designed the experiments, analyzed the data, wrote the paper.
- Ismael A. Delgado-Leal, Misael J. Reyes-Rodríguez and César V. Elizondo-Solis performed the experiments, contributed reagents/materials/analysis tools, wrote the paper, prepared figures and/or tables.
- Gerardo E. Lozano-Luévano performed the experiments, analyzed the data, contributed reagents/materials/analysis tools, prepared figures and/or tables.
- Teresa A. Nava-Obregón analyzed the data, contributed reagents/materials/analysis tools.
- Dionicio Palacios-Ríos conceived and designed the experiments, analyzed the data, reviewed drafts of the paper.

### Human Ethics

The following information was supplied relating to ethical approvals (i.e., approving body and any reference numbers):

Ethical committee of the Faculty of Medicine of the Autonomous University of Nuevo León: AN15-011.

### Data Availability

The raw data has been supplied as Data S1.

### Supplemental Information

Supplemental information for this article can be found online at http://dx.doi.org/10.7717/peerj.2255#supplemental-information.

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
