# Peer review of "Perception of the risk of adverse reactions to analgesics: differences between medical students and residents"

_PeerJ, doi:10.7717/peerj.2255_

## Round 0.1 · original submission · Major Revisions

· Academic Editor

Major Revisions

Dear Dr Castillo-Guzman,

Your manuscript has been reviewed by two expert reviewers and they have identified a number of issues that need to be addressed in the revised manuscript. Please provide point-by-point response to reviewers' comments.

Reviewer 1 ·

Basic reporting

Generally well written and presented. A final proof to ensure clear English throughout will be worthwhile.
Clarity of table 1 may be improved by presenting in landscape format.

Minor point - line 58 should either specify "vitamin k antagonists" or should generalise to the broader notion of bleeding risk, rather than INR (consider novel anticoagulants).

Experimental design

Research question - could be clarified - line 67 states the aim is to "investigate". The work presented assesses the difference in perception by gender and level of qualification.
Methods -
Suggest comment on a sample size calculation (or need for one).
Can you clarify how you estimated the overall risk score for each drug class - was it the sum of the score for each of the six ADRs or was it presented as a separate item?
Suggest clarify methods (line 76) - were ADRs or perception of risk of ADRs assessed?
Suggest clarify the method of data collection - line 76 states "interviewed", was this self-administered, was there any potential bias associated with interview, what was the response rate?
How were NSAIDs defined?

Discussion, lines 140-143 would benefit from a supporting citation.

Validity of the findings

Table 2 - why are results presented for only two of the four drug classes investigated?
Tables 1 and 2 could benefit from inclusion of the p-value and the level of significance for both row and column - it is not apparent in these tables whether there was a difference in risk perception by gender.

Suggest consider relating the findings to the evidence regarding development of prescribing competence in medical students and junior doctors; and clinical, experiential training. This may be relevant to the differences identified between medical students and residents.
The authors have assessed the difference in risk perception by gender, but have not commented on this in the discussion.
Can the authors comment on the finding that NSAIDs were identified as having addictive potential.

(The data do not appear to have been provided).

Conclusions - final two sentences do not seem to be supported by the findings of this study.

Reviewer 2 ·

Basic reporting

This topic is an interesting one, and worthy of study. The authors have provided appropriate context for their research and have demonstrated how this work fits into the broader field of knowledge, building on the existing literature. They have cited relevant papers, and in general they are referenced appropriately although some deficiencies have been noted which could be addressed by more thorough proof-reading (e.g. some repetition of page numbers in lines 176, 183; an incomplete title in line 190; inadequate details in line 204 [DOI or URL would be expected for an online reference since merely providing the volume and issue number is not sufficiently precise]). There are also some unsupported statements that require references – e.g. paracetamol ‘is the single most important cause of acute fulminant hepatic failure’ (lines 125-126); ‘studies suggest that the increase of information to patients will lead to a reduction in ADR’ (lines 141-142).

The paper would also benefit from a specific linguistic review to address multiple deficiencies in spelling, syntax and manner of expression, including in the figure and table captions.

The figure and tables have potential to contribute towards the paper’s value. However, there are aspects requiring improvement:

Figure 1:
• The y-axis is untitled.
• There is no indication of what the data represent (e.g. mean +/- standard deviation or standard error; median +/- interquartile range).
• It would be preferable if the x-axis were simply labelled with ‘medical students’ and ‘residents’ rather than the current codes 0 and 1 with a key explaining their meaning.

Tables 1 and 2:
In both tables -
• There is no indication of what the data represent (means, medians etc.).
• No measure of spread (e.g. SD or IQR) has been provided.
• N should be specified for the various groups and subgroups.
• It is not sufficiently clear what groups have been compared for each statement of significance. For example, in Table 1 does the ‘S’ (significant) in the row for males and section for paracetamol represent a significant difference between the scores of male residents and male medical students for this drug (presumably the case), or between males and females? More precise captions would remove any ambiguity.
• The significance level should be expressed as ‘P < 0.05’ (not ‘P = < 0.05’).
• The statistical test used should be named in each table so they are self-contained.
In addition, for Table 1 -
• The table needs to be widened or the font size reduced, so that numbers with two decimal places are not split over two lines (in the paracetamol, morphine and NSAID sections).
In addition, for Table 2 –
• No rationale has been provided for including only morphine and NSAIDs in this table. Why not also paracetamol and COX-2 inhibitors?

The raw data provided do not appear to be complete. While the headings in the Excel file are not comprehensive, it appears as though individual participants’ ratings for only two drug types have been included (green and yellow sections) rather than four.

Experimental design

The research appears to be original in the specific groups examined, although very closely modelled on previous work in the field, and it is within the journal’s scope. Ethical approval was obtained.

The researchers appear to have had a clear research question in mind relating to comparison of ‘risk perception’ for different groups of participants and drugs. However, since the specific directions given to participants are not included anywhere in the paper, it is not clear what exactly is being assessed:
- Perception of how *frequently* certain adverse effects occur with various drugs?
- Perception of the *severity* of harm (from certain adverse effects) likely to arise with various drugs?
- Some kind of *general* perception of risk (from certain adverse effects) derived from combining frequency and severity (and potentially other factors)?
- Whether the assessment relates to normal use or potential misuse/excessive use? (Or whether indeed the participants were given any directions on this. Lack of guidance would comprise a limitation requiring acknowledgement as participants may have interpreted the task in different ways.)

Furthermore, having obtained risk ratings for individual potential adverse effects (hepatotoxicity, sedation etc.) for each analgesic, it is not clear how these were used to derive the overall risk rating for each analgesic, or whether the participants were asked to give a separate global risk rating.

The quality of the research cannot be properly judged without further information on the experimental design – in particular the specific question(s) posed to participants. The lack of detail obviously also detracts from the potential to replicate the study.

There is no mention in the methodology of randomization of the drug order for individual participants so presumably the questionnaire(?) [line 87] or interview(?) [line 90] was conducted with the same sequence of questions for all participants. This should be considered as a limitation, since the order may have influenced responses – e.g. asking about the safety of NSAIDs directly after paracetamol may have led to different results than asking about NSAID safety after morphine safety.

It is also not clear precisely how the data collection tool was administered: Self-completion under observation, since a visual analogue scale is described? Interview as mentioned in line 90?

Was the study anonymous? True anonymity (or lack thereof) may have affected participation and hence the outcomes (e.g. only conscientious students or those confident in their knowledge might have chosen to participate). If the study was undertaken by interview, true anonymity would require not just non-recording of names, but conduct of the interview by an external researcher with no possibility of acquaintance. Information on anonymity should be included and if the study was not fully anonymous this should be acknowledged as a limitation.

What year(s) were the medical students drawn from? The manuscript mentions (lines 73-74) that they were ‘those who had already taken a pharmacology course’ and the discussion alludes (lines 118-119) to ‘recent courses of pharmacology’. However, there is no indication of whether the pharmacology course mentioned in line 74 related to analgesia or not, nor is there an indication of how close in time the research study was to the pharmacology education. The fact that there were 505 participants suggests they were drawn from multiple years of the course. However, previous longitudinal research on a similar topic (Durrieu et al., doi: 10.1111/j.1472-8206.2009.00783.x) suggests that with time and some clinical experience (which students in higher years of the course presumably would have), perceptions of the adverse effect risk associated with various medicines can change, so the responses of students in second or third year might be expected to differ from those in fifth year. Hence if students from multiple years were included it would be desirable to stratify the findings or acknowledge this as a limitation.

As already noted, the figure/tables do not indicate the nature of the summary data (means, medians etc.). Similarly, the nature of the numbers provided in the text of the results section is not explicitly stated. However, the methods (line 82) mention means, 25th-75th centiles and the Mann-Whitney U-test. If centiles/quartiles and a non-parametric test such as Mann-Whitney are being used, the median rather than mean would be expected as the central measure. It would be helpful if the authors had explored the normality of the data, as this would guide the appropriate approach.

Validity of the findings

Without the additional methodological information noted above, it is difficult to judge the robustness of the data. Many of the comments above are therefore also relevant to this section.

Once the exact questions posed to participants have been clarified, it will be important to ensure that the authors do not make findings beyond the scope of the data. For example, much of the results and discussion suggest that risk classification focused on the frequency of ADR occurrence (e.g. line 98: ‘most and least probable adverse effects’; lines 135-136: ‘GI bleeding was identified as the most common’). However, elsewhere the authors focus on both severity and frequency (e.g. line 100: ‘a major risk more often’), while in many places the terminology is ambiguous. It is likely that at least some of these statements are not justified by the data collected.

No concrete statistics (e.g. incidence rates) for the actual (measured) risks associated with analgesics have been provided from the literature. This is a significant omission since the authors seek to infer morphinophobia in both groups of participants, and a need for ongoing physician education on ADRs post-qualification. Neither of these conclusions can explicitly be drawn from the data in the absence of information on how realistic the participants’ assessments of morphine’s risks were, and the comparative accuracy of risk estimation for the two groups of participants.

The authors state (lines 124-126) that ‘The low risk perceived for paracetamol could have serious implications. Clearly they [participants] underestimated its risk in spite of [it] being the single most important cause of acute fulminant hepatic failure.’ However this conclusion cannot be drawn and the latter part of the sentence (unreferenced) is irrelevant without considering the frequency with which acute fulminant hepatic failure occurs, if the study’s risk classification incorporates a frequency component as it seems to have done. Moreover, the question of appropriate and inappropriate dosage also arises in this context: The authors earlier acknowledged (lines 49-50) that ‘paracetamol is relatively safe when taken in a therapeutic dose’. Therefore we again return to the question of whether participants were guided on whether to consider the risks of conventional therapeutic doses or in overdose.

Four drugs/drug classes were investigated. However, in various subsections the results and discussion focus on two or three analgesics only, and the rationale for this is not always clear.

It seems likely that the medical students were drawn from multiple years of the degree programme. As mentioned above, this is a concern in light of the variation in their level of education and experience, and undermines the validity of conclusions drawn from pooled data.

As already noted, the raw data file provided appears to be incomplete.

Comments for the author

While this research has potential value, there is too much information missing, and too many unjustified statements made, to warrant publication of the current manuscript.

---

## Round 0.2 · accepted · Accept

· Academic Editor

Accept

Dear Sandra,

Thank you for your submission to PeerJ, which I am happy to Accept.